# A Rare Case of Multiple Gastrointestinal Stromal Tumors Coexisting with a Rectal Adenocarcinoma in a Patient with Attenuated Familial Adenomatous Polyposis Syndrome and a Mini Review of the Literature

**DOI:** 10.3390/medicina58081116

**Published:** 2022-08-18

**Authors:** Daniel Paramythiotis, Filippos Kyriakidis, Eleni Karlafti, Triantafyllia Koletsa, Anastasia Tsakona, Petros Papalexis, Aristeidis Ioannidis, Petra Malliou, Smaro Netta, Antonios Michalopoulos

**Affiliations:** 1First Propaedeutic Surgery Department, AHEPA University General Hospital of Thessaloniki, 54621 Thessaloniki, Greece; 2Second Chemotherapy Department, Theagenio Cancer Hospital of Thessaloniki, 54639 Thessaloniki, Greece; 3Emergency Department, AHEPA General University Hospital, Aristotle University of Thessaloniki, 54621 Thessaloniki, Greece; 41st Propaedeutic Department of Internal Medicine, University General Hospital of Thessaloniki AHEPA, Aristotle University of Thessaloniki, 54621 Thessaloniki, Greece; 5Pathology Department, Faculty of Medicine, Aristotle University of Thessaloniki, 54621 Thessaloniki, Greece; 6Unit of Endocrinology, First Department of Internal Medicine, Laiko General Hospital, National and Kapodistrian University of Athens, 11527 Athens, Greece

**Keywords:** multiple GIST, AFAP, rectal, adenocarcinoma, surgery, case report

## Abstract

Background: Multiple gastrointestinal stromal tumors (GISTs) are extremely rare entities that exist either as spontaneous GISTs or as part of various syndromes, such as Carney’s triad and type I neurofibromatosis (NF1). Attenuated familial adenomatous polyposis (AFAP) is a variant of familial adenomatous polyposis (FAP) with a milder clinical presentation. Both GISTs and AFAP have been reported to coexist with colorectal cancer, but the coexistence of GISTs and AFAP has never been reported in the literature before. Case report: A 45-year-old male patient with known AFAP arrived scheduled for a total colectomy and ileo-rectal anastomosis due to the malignancy of one of the previously biopsied polyps of the upper rectum. Intraoperatively, multiple nodular tumors were found at the jejunum within a length of 45 cm, for which an enterectomy and enteroanastomosis were performed. A histopathological examination of the whole colectomy specimen confirmed the presence of multiple polyps in the large intestine along with a rectal invasive adenocarcinoma. At the same time, in the examined part of the small intestine, 15 GISTs sized from 0.5 to 2.0 cm of prognostic group I, were identified. The patient’s postoperative course was uncomplicated. Conclusion: Multiple GISTs may present as an asymptomatic disease, and the same thing is true for colorectal cancer. Therefore, the appropriate screening is crucial for entities such as AFAP, since the surgery was performed because of the malignant transformation in one of the polyps and revealed multiple GISTs, as well.

## 1. Introduction

GISTs represent the most common neoplasm of mesenchymal origin of the gastrointestinal tract, with an incidence of approximately 15 cases per million per year. The distribution, based on sex, shows that men are affected slightly more often than women [1]. Their symptoms vary depending on their site, and they include abdominal pain and discomfort, early satiety, gastrointestinal (GI) bleeding, and anemia-related fatigue [2]. Mutations that lead to the activation of KIT and PDGFRA genes are the basis for the development of these tumors in most cases. Almost 80% of GISTs have a mutation regarding the KIT gene, whereas this percentage for PDGFRA is 5–10% [3]. Thus, there are 10–15% of GISTs that have neither of these mutations (wild-type GISTs). The most common localizations of GISTs are in the stomach (60%) and small intestine (30%), although it is possible to find them anywhere across the gastrointestinal tract [4].

The morphology of the cells and the findings of immunohistochemistry are the major factors for the pathological diagnosis of GIST. Regarding morphology, there are three types: spindle cell type (70%), epithelioid cell type (20%), and mixed type (10%) [5]. Furthermore, 95% and 70% of GISTs show positivity in immunohistochemistry for KIT (CD117) and CD34, respectively [6]. Even though KIT positivity alone is not diagnostic, it is a feature with great importance for the diagnosis of a GIST if a tumor’s morphological features are compatible with GISTs [2]. In the remaining percentage of GISTs where the immunohistochemistry for KIT is negative, DOG1 staining, followed by CD34 staining, is considered diagnostic [5].

Regarding GISTs, there are four major independent prognostic factors: the mitotic index, tumor location (gastric vs. non-gastric), tumor size, and tumor rapture. After complete resection, surgery alone is enough for approximately 60% of GIST patients; however, for the remaining 40%, additional targeted therapy is required due to relapses [7].

The occurrence of multiple GISTs is extremely rare, accounting for about 2% of all GISTs according to one study [8], although this number may not be precise due to their very low incidence. In most cases, multiple GISTs are associated with specific syndromes, such as Carney’s triad and type I neurofibromatosis, even though there are also spontaneous ones [9]. Nonetheless, according to our research, there is no other published case with the combination of AFAP with multiple GISTs.

## 2. Case Presentation

A 45-year-old male patient with known AFAP arrived scheduled for a total colectomy and ileo-rectal anastomosis due to the malignancy of one of the previously biopsied polyps of the upper rectum. During operation, multiple nodular tumors were found at the beginning of the jejunum, about 10 cm below the ligament of Treiz (Figure 1) and within a length of 45 cm. These tumors were proven to be of mesenchymal origin in intraoperative frozen sections, with features compatible with GISTs, and an enterectomy and enteroanastomosis were performed. It is worth mentioning that a gastroduodenoscopy had previously been performed, and no polyps were revealed.

Results: Gross examination of the whole colectomy specimen revealed multiple polyps in the large intestine, which ranged in size from 0.3 to 2 cm, and a rectal adenocarcinoma that measured 3.5 cm at its greatest diameter. The latter, on histological grounds, corresponded to invasive low-grade colorectal adenocarcinoma, NOS, stage pT2N2a according to AJCC, 8th ed. In addition, one of the polyps showed invasion limited to the polyp head—level 1, based on the Haggitt classification.

Gross examination of the small intestine specimen showed 15 nodules located on the muscularis externa and serosa, measuring from 0.5 to 2.0 cm at their greatest diameter. Some of them were narrowed (Figure 2A). Hematoxylin and eosin-stained sections from the nodules showed spindle cells with pale eosinophilic cytoplasm and elongated nuclei with inconspicuous nucleoli (Figure 2B). Mitoses were scarce, at up to 1 per 50 high power fields (HPF). Necroses and hemorrhages were absent. Upon immunohistochemical study, the neoplastic spindle cells were positive for CD34, CD117 (Figure 2C), and DOG-1 (Figure 2D) antibodies. A few cells were also positive for smooth muscle actin (SMA), whereas immunostains for S-100 and desmin were negative. Ki67 exhibited positivity in almost 2% of neoplastic cells. The findings set the diagnosis of multiple gastrointestinal stromal tumors of prognostic group I.

The patient’s postoperative course was uncomplicated.

## 3. Discussion

Attenuated familial polyposis (AFAP), is a phenotypic variant of FAP syndrome, characterized by the presence of 10 to 100 synchronous polyps in the large intestine. In classical FAP, there are thousands of polyps, which is why this form is called “attenuated”. AFAP is inherited in an autosomal dominant manner, but APC gene mutations are different from those detected in patients with classic FAP, since the mutations in APC associated with AFAP are mostly detected in three regions of the gene: the 5′ end (the first 5 exons), exon 9, and the distal 3′ end [10,11]. In individuals with mutations in region 1, the average number of adenomas is, in general, higher and more variable compared to individuals with mutations in regions 2 and 3 [12]. AFAP is generally managed with regular screening to detect if and when polyps develop. Screening by colonoscopy has been recommended for affected people starting at age 20 to 25 years. Because individuals with AFAP can also develop duodenal adenomas and other cancers, upper endoscopy is typically recommended starting at age 20 to 30 years, and then continuing every one to three years depending on the number of polyps. The incidence of desmoid tumors (DT) in classic FAP is approximately 10%, but it seems to be lower in AFAP patients [10]. There is also an increased risk of developing colorectal cancer compared to the general population, although the risk is lower compared to FAP patients [13]. In this presenting case, AFAP was confirmed by NGS analysis. Interestingly, numerous GISTs were found in the jejunum. A thorough search in the PubMed database revealed no other published case with the coexistence of attenuated familial polyposis and multiple GISTs. The combination of these two entities (AFAP and GISTs) is extremely rare and is a challenge to deal with properly.

According to a meta-analysis, there is a risk for GIST survivors to develop second primary tumors (SPTs) [14]. In this study, it was found that the most common sites of GISTs associated with SPTs were the gastro-esophageal (67.2%) and small intestinal (19.8%) sites. On the other hand, regarding the incidence of SPTs according to their anatomic sites, the most common were the colorectal (17%) and prostate (14%) sites.

Regarding genotyping for GISTs, there is a recommendation for cases in which the plan includes medical treatment [2]. Since this was not the case in our patient, a genetic analysis of the tumor was not performed, so, unfortunately, this information is unavailable. Nonetheless, the importance of genotyping should not be underestimated when it is indicated because of the valuable data it provides. Tumor genotype has been proposed as an additional prognostic factor. More specifically, associations between worse DFS (disease-free survival) and KIT exon 9 duplications or KIT exon 11 deletions have been observed, while, on the other side, PDGFRA exon 18 mutations seem to be correlated with better prognosis [15].

Another important aspect is how the mutational status affects the response to treatment. For patients treated with standard-dose imatinib, OS (overall survival) and PFS (progression-free survival) were higher in those with KIT exon 11 mutations compared to those with KIT exon 9 mutations or with wild-type GISTs [16]. On the other hand, in imatinib-resistant cases of GIST treated with sunitinib, response rates, PFS, and OS were better in patients with primary KIT exon 9 mutations or with wild-type GISTs in comparison to those with KIT exon 11 mutations [17]. For the advanced PDGFRA mutant GISTs, the data show that patients with D842V substitution in exon 18 are resistant, while patients with other mutations in the gene seem to be sensitive to imatinib [18].

Concerning multiple GISTs, there are two different categories—(a) spontaneous and sporadic (b), where the emerging lesions are associated with specific syndromes (the most common being Carney’s triad and NF1) [9].

NF1 is an autosomal dominant disease that has, as major characteristics, neurofibromas and café-au-lait spots and is caused by germline NF1 mutations [19]. There is a well-known association between GISTs and NF1, and there is a 7% risk for patients with NF1 to develop a GIST [20]. Most NF1-associated GISTs are less aggressive regarding their clinical and pathological features, present as numerous small, asymptomatic lesions in the small intestine, in most cases, and, rarely, are KIT or PDGFRA mutants [5,9].

There are some published cases where GISTs have been found to coexist with colorectal adenocarcinomas in the ground of NF1 [21].

The Carney triad is a non-heritable syndrome that consists of multiple gastric GISTs, paragangliomas, and pulmonary chondroma [22]. The mechanism of tumor development includes an epigenetic modification where succinate dehydrogenase C (SDHC) is inactivated through hypermethylation [23]. GISTs in the Carney triad are negative for KIT and PDGFRA mutations [9].

Multiple sporadic GISTs are rarely described, but it is worth mentioning that the prognosis of patients with multiple GISTs (whether they are spontaneous or developed within the basis of a syndrome) is the same as that of patients with solitary tumors [9].

As far as it concerns the association of these pathologies, in our case, there are some possible explanations. In the first scenario, the rectal adenocarcinoma could be associated with the multiple GIST, since this is something described in the literature [14] and the AFAP was a coexisting pathology. In the second scenario, the rectal adenocarcinoma was developed within the basis of the underlying AFAP [13], and the multiple GISTs were a coexisting pathology. There is no direct connection described in the literature between AFAP and multiple GISTs.

Regarding the type of operation performed on our patient, it is worth mentioning that an interesting alternative is a perineal colostomy, which is a type of colostomy that aims to make use of the natural anal orifice as the ostomy’s point of exit while restoring a sphincteric function. This procedure has a lot of advantages, some of them being the higher performance of the patients on everyday functionality scores, satisfactory continence, sufficient sphincteric function, and better quality of sexual life [24]. Despite all these, this procedure was not feasible for our patient, as a total colectomy was performed, making a colostomy not possible. The choice of performing the total colectomy was based upon the underlying AFAP, and, since there were no polyps in the peripheral stump of the ileo-rectal anastomosis, the aim was to have no residual disease.

### Staging Based Approach

Stage I, according to the latest NCCN guidelines, includes GISTs that are T1 (tumor < 2 cm) or T2 (tumor between 2 cm and 5 cm) N0 M0 and have a low mitotic rate (5 or fewer mitoses per mm or per 50 HPF) [2]. In our case, all GISTs were 0.5–2 cm and had up to 1 mitosis per 50 HPF, with no nodes or metastases observed. Therefore, all were T1N0M0 with a low mitotic rate and were classified as stage I. Since they were also completely resected, they are considered low risk with a very good prognosis, and the most optimal approach is observation after surgery [2,25]. For patients with completely resected GISTs with no preoperative imatinib, adjuvant imatinib is preferred only for intermediate and high-risk patients [2,26,27].

Regarding rectal adenocarcinoma, according to the latest NCCN guidelines [28], the tumor is categorized as T2, since it invades the muscularis propria and does not invade the pericolorectal tissues; N2a, since 4–6 lymph nodes were found to be positive (in our case, 4 nodes were found to be positive); and M0, since there was no evidence of distant metastasis.

For tumors that, after transabdominal resection, are found to be pT1-2, N1-2 M0, the approach is the administration of adjuvant chemotherapy and radiotherapy (RT) [28]. More specifically, the suggestion is chemotherapy with FOLFOX (Folinic acid + 5-Fluorouracil (5-FU) + oxaliplatin) or CAPEOX (capecitabine + oxaliplatin), followed by infusional 5-FU or capecitabine + RT, or vice versa, with the administration of chemoradiotherapy being first followed by FOLFOX or CAPEOX [29,30,31].

## 4. Conclusions

It cannot be certain whether this tumor developed within the basis of a syndrome associated both with the development of GISTs and other primary tumors or within the basis of the AFAP, with which the patient was diagnosed years ago. Further details regarding molecular, genetic, and immunohistochemical analysis would be needed to support either one of these theories. In any case, the screening of the patient due to the underlying AFAP was crucial for the detection of the rectal tumor, which led to the decision to perform surgery and finally the revelation of the multiple GISTs.

## Figures and Tables

**Figure 1 medicina-58-01116-f001:**
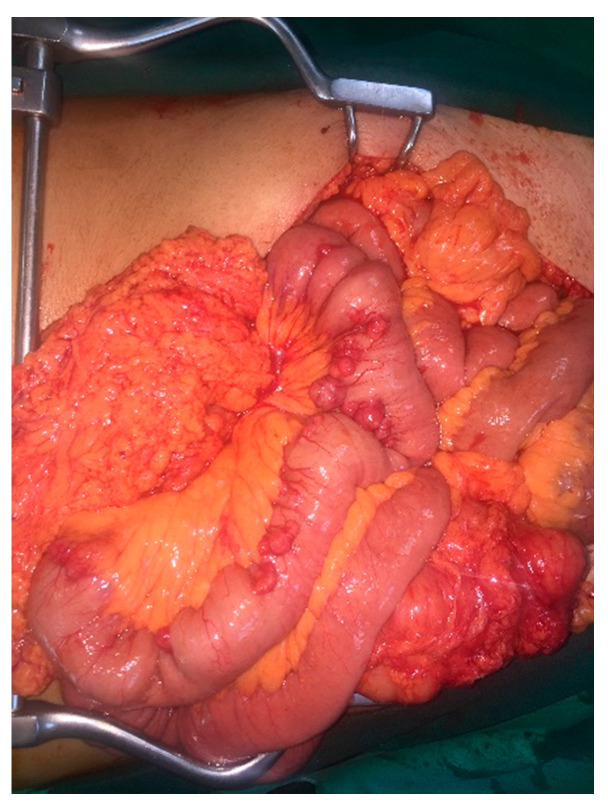
Image of the GISTs during the surgery.

**Figure 2 medicina-58-01116-f002:**
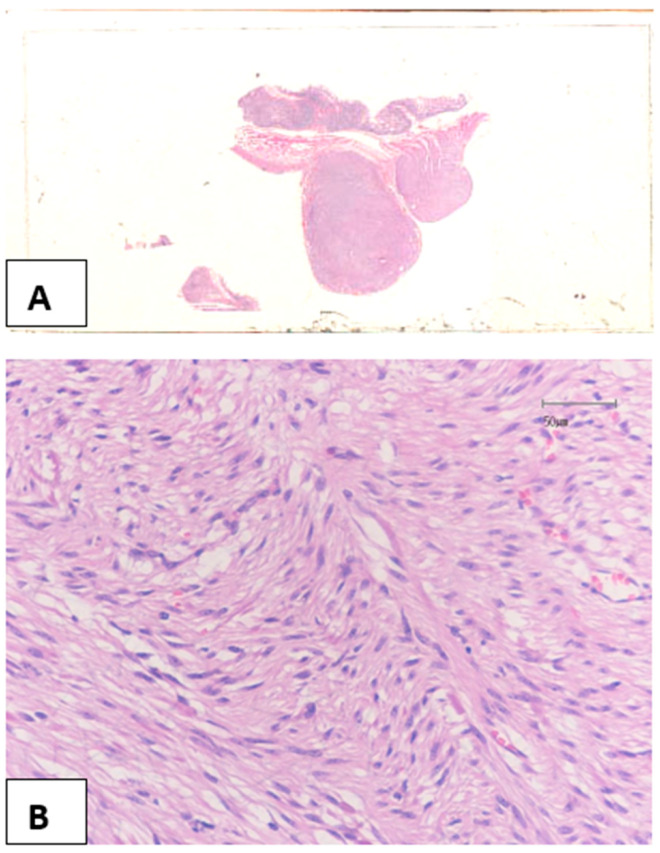
Two nodular neoplasms close to each other (**A**), consisting of spindle cells without significant atypia or pleomorphism (**B**), proven to be GISTs positive for CD117 (**C**) and DOG-1 (**D**) antibodies, ((**B**) Hematoxylin-Eosin X200; (**C**,**D**) Immunohistochemistry X200).

## Data Availability

PubMed was used as a source of information.

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
