# Peer review of "A Rare Case of Multiple Gastrointestinal Stromal Tumors Coexisting with a Rectal Adenocarcinoma in a Patient with Attenuated Familial Adenomatous Polyposis Syndrome and a Mini Review of the Literature"

_medicina, 2022, doi:10.3390/medicina58081116_

Round 1

Reviewer 1 Report

I would like to congratulate the authors on their fascinating work regarding this very interesting case report. The manuscript is well-written and the incorporated figures make the study easy to follow.

A perineal stoma is also an operation preferred for patients with attenuated familial adenomatous polyposis. I would like a brief discussion on this procedure and consider citing the recently published article:

https://pubmed.ncbi.nlm.nih.gov/35664027/

It can be accepted for publication after this minor revision.

Author Response

We would like to thank you for your comments. We followed your recomendations and we added a paragraph in the discussion, regarding the topic of perineal colostomy.

Reviewer 2 Report

This is a potentially interesting case presentation about a rare associated pathology. The case is well presented and nicely illustrated. However, few concerns and comments should be raised:

The Discussion part should include few potential explanations for this rare association of pathologies. Otherwise, it is just a random association with no clinical meaning.

The manuscript should be revised to correct few minor errors. For example, in line 25 please replace “rear” with “rare”; in line 127 please replace “jujenum” with “jejunum” etc.

The quality of Figure 2A is not so good. Please try to improve it.

Author Response

We would like to thank you for your comments.

We followed your recomendations and added a paragraph in the discussion section,regarding the association of these pathologies and also added a statement in the conclusion so as to be better supported by the results.

We corrected hopefully all the minor errors.

Regarding the quality of the Figure 2A,despite our efforts this was the best we could achieve.

Round 2

Reviewer 2 Report

The authors addressed all the concerns raised by the reviewers

Author Response

Thank you very much for your comments!